# One-particle density matrix of trapped one-dimensional impenetrable bosons from conformal invariance

**Yannis Brun*** and **Jérôme Dubail†**

CNRS & IJL-UMR 7198, Université de Lorraine, F-54506 Vandoeuvre-lès-Nancy, France

* yannis.brun@univ-lorraine.fr
† jerome.dubail@univ-lorraine.fr

## Abstract

The one-particle density matrix of the one-dimensional Tonks-Girardeau gas with inhomogeneous density profile is calculated, thanks to a recent observation that relates this system to a two-dimensional conformal field theory in curved space. The result is asymptotically exact in the limit of large particle density and small density variation, and holds for arbitrary trapping potentials. In the particular case of a harmonic trap, we recover a formula obtained by Forrester et al. [23] from a different method.



# 1 Introduction

In the past decade, the one-dimensional (1d) gas of bosons with delta repulsion—solved by Lieb and Liniger in 1963 [1] (see also Ref. [2])—has played a central role in the modeling of the series of groundbreaking ultracold atom experiments that have lead to isolated 1d quantum systems [3–8]; see Refs. [9–12] for an overview of the state of the art on the theory side, and discussions of its experimental relevance. The hamiltonian of the 1d delta Bose gas is, in second-quantized form,

$$H = \int dx \left[ \Psi^\dagger \left( -\frac{\hbar^2}{2m}\partial_x^2 - \mu + V(x) \right)\Psi + \frac{g}{2}\Psi^{\dagger 2}\Psi^2 \right], \tag{1}$$

where $\Psi^\dagger(x)$ and $\Psi(x)$ are the boson creation/annihilation operators that obey the canonical commutation relation $\left[\Psi^\dagger(x), \Psi(x')\right] = \delta(x-x')$. $m$ is the mass of a particle, $\mu$ is the chemical potential, $V(x)$ is a confining potential, and $g > 0$ is the interaction strength. In this paper, we focus on the Tonks-Girardeau limit [13],

$$g \to +\infty\,, \tag{2}$$

where the bosons become impenetrable. One quantity that has been studied regularly since the 1960s in this model of impenetrable bosons (or Tonks-Girardeau gas) is the *one-particle density matrix,* see e.g. [14–28]. It is defined as

$$g_1(x,x') = \langle\psi|\Psi^\dagger(x)\Psi(x')|\psi\rangle\,, \tag{3}$$

where $|\psi\rangle$ is the ground state of the Hamiltonian (1). The initial motivation for studying this one-particle density matrix came from its role in defining a criterion for Bose-Einstein condensation [29]: when the eigenvalues of $g_1(x,x')$ are viewed as occupation numbers for the interacting bosons, Bose-Einstein condensation is reached if the largest of these occupation numbers is of order $O(N)$ when the number of particles $N$ goes to infinity. Schultz pointed out in 1963 [14] that one-dimensional impenetrable bosons always fail to fulfill this criterion. The one-particle density matrix of the Tonks-Girardeau gas also appeared in the mathematics literature, in connection with random matrix theory and Toeplitz determinants, see e.g. [15–18, 23, 26, 30–32].

For the convenience of the reader, we give a brief summary of the known results on $g_1(x,x')$ that are motivating this paper, see Table 1. An instructive discussion of those results can also be found in the review [11]. In 1964, Lenard [15] showed that, in the translation-invariant Tonks-Girardeau gas with uniform density $\rho$, $g_1(x,x')$ could be written as a Toeplitz determinant, and used this to derive a bound on the asymptotic behavior for $|x-x'| \gg \rho^{-1}$: he managed to show that, in this regime, $g_1(x,x') \le A/|x-x'|^{\frac{1}{2}}$ for some constant $A > 0$. He came back to this result in 1972 [30], giving a better estimate for the constant $A$, and conjectured that the upper bound was actually an equality in the limit $\rho|x-x'| \to \infty$; this result was proved by Widom in 1973 [31]. The behavior of $g_1(x,x')$ at large $|x-x'|$ in an infinitely long system thus follows from the results of those three references; however, it seems to have been written down explicitly only in 1979 by Vaidya and Tracy [16],

$$g_1(x,x') = |C|^2 \frac{\rho^{\frac{1}{2}}}{|x-x'|^{\frac{1}{2}}}, \qquad \text{when} \quad |x-x'| \gg \rho^{-1}. \tag{4}$$

The numerical constant $|C|^2$ is known exactly, and it can be expressed in terms of Barnes' G-function $G(.)$,

$$|C|^2 = \frac{G^4(3/2)}{\sqrt{2\pi}} \simeq 0.521414\,. \tag{5}$$

| Geometry of the system | Density matrix $g_1(x, x')$ |
|---|---|
| Infinite line, uniform density $\rho$ <br><br> ![line diagram with points $x$ and $x'$] | $\dfrac{G^4(3/2)}{\sqrt{2\pi}} \dfrac{\rho^{\frac{1}{2}}}{\|x - x'\|^{\frac{1}{2}}}$ <br><br> Ref. [16] and Refs. [15, 30, 31] |
| Periodic system, uniform density $\rho$ <br><br> ![circle diagram with $L$, $x$, $x'$] | $\dfrac{G^4(3/2)}{\sqrt{2\pi}} \dfrac{\rho^{\frac{1}{2}}}{\left\| \frac{L}{\pi} \sin \frac{\pi}{L}(x - x') \right\|^{\frac{1}{2}}}$ <br><br> Refs. [15, 30, 31], quoted in Ref. [23] |
| Harmonic trap, $\langle \rho(x) \rangle = \frac{1}{\pi\hbar}\sqrt{2m(\mu - \frac{m\omega^2 x^2}{2})}$ <br><br> $V(x) - \mu$ <br> ![parabola with $x$, $x'$, $x_1$, $x_2$] | $\dfrac{G^4(3/2)}{\sqrt{2\pi}} \dfrac{\langle \rho(x) \rangle^{\frac{1}{4}} \langle \rho(x') \rangle^{\frac{1}{4}}}{\|x - x'\|^{\frac{1}{2}}}$ <br><br> Refs. [23, 24] |
| Arbitrary trap potential <br><br> $V(x) - \mu$ <br> ![double-well potential with $x$, $x'$, $x_1$, $x_2$] | $\dfrac{G^4(3/2)}{\sqrt{2\pi}} \sqrt{\dfrac{m}{2\hbar \tilde{L}}} \dfrac{\left\| \sin\left(\frac{\pi}{\tilde{L}}\tilde{x}\right) \right\|^{\frac{1}{4}} \left\| \sin\left(\frac{\pi}{\tilde{L}}\tilde{x}'\right) \right\|^{\frac{1}{4}}}{\left\| \sin\left(\frac{\pi}{\tilde{L}}\frac{\tilde{x}-\tilde{x}'}{2}\right) \right\|^{\frac{1}{2}} \left\| \sin\left(\frac{\pi}{\tilde{L}}\frac{\tilde{x}+\tilde{x}'}{2}\right) \right\|^{\frac{1}{2}}}$ <br><br> The result of this paper. |

Table 1: One-particle density matrix $g_1(x, x')$ of the Tonks-Girardeau gas for different geometries. The first three lines correspond to known results in the literature. The fourth line is the main result of this paper, with $\tilde{x}$ defined in Eq. (8).

Vaidya and Tracy also studied systematically the higher order terms that appear at finite $\rho|x - x'|$ in (4); see also Jimbo et al. [17] and Gangardt [24] about these terms. The asymptotic result analogous to (4) for a periodic system of length $L$ also follows from Refs. [15, 30, 31], and is quoted in Ref. [23]. It is given in the second row of Table 1.

Then, it should be emphasized that it took almost 25 years to go from the translation-invariant case to the harmonically trapped system, $V(x) = \frac{1}{2}m\omega^2 x^2$, and to reach the formula given in the third row of Table 1. It was cracked only in 2003 by Forrester, Frankel, Garoni and Witte [23], with methods coming from random matrix theory (see also Gangardt [24], who used a different method). We stress that, for a long time, it was by no means obvious how to deal with a system with inhomogeneous density, and it required rather advanced techniques to deal with the harmonic trap. These techniques are specific to the harmonic potential, and it seems difficult to generalize them to more general inhomogeneous situations.

In this paper, we extend this long series of studies of $g_1(x, x')$ by providing an asymptoti-

cally exact formula for arbitrary confining potentials $V(x)$. To do this, we rely on simple ideas of 2d conformal field theory (CFT) [33]. We use the recent results of Refs. [34, 35], which show that it is possible to use CFT techniques even when the system is inhomogeneous in the bulk. In particular, the Tonks-Girardeau gas in a trap potential is in fact related to a CFT in curved 2d space [35]; this is the main insight that we use in this paper (we will come back to this in section 3). As usual with CFT, the results are exact, but not mathematically rigorous.

In order to clarify the domain of validity of our main result, we detail here the setup used throughout the paper. First and foremost, we always focus on the semiclassical limit,

$$\hbar \to 0, \qquad \text{with} \quad m, \mu, V(x) \quad \text{fixed.} \qquad (6)$$

In that limit, the local density approximation (LDA) becomes exact, and the density of particles at point $x$ is given by

$$\langle \rho(x) \rangle = \langle \psi | \Psi^\dagger(x) \Psi(x) | \psi \rangle = \begin{cases} \frac{1}{\pi \hbar} \sqrt{2m(\mu - V(x))} & \text{if } \mu - V(x) > 0, \\ 0 & \text{otherwise.} \end{cases} \qquad (7)$$

We assume that the region where $\mu - V(x) > 0$ is a single non-empty interval $[x_1, x_2]$. For any position $x$ inside the interval, we introduce $\tilde{x}$ as

$$\tilde{x} = \int_{x_1}^{x} \frac{du}{v(u)}, \qquad \text{where} \quad v(x) = \frac{\pi \hbar}{m} \langle \rho(x) \rangle, \qquad (8)$$

so that $\tilde{x}$ represents the time needed by a signal emitted from the left boundary of the system $x_1$ to reach the point $x$, given that it propagates with velocity $v(x)$. We also call $\tilde{L} = \int_{x_1}^{x_2} \frac{du}{v(u)}$ the time needed by a signal to travel from one boundary to the other. The limiting procedure (6) is then nothing but a particularly convenient way of taking the thermodynamic limit $N \to \infty$, since the total number of particles in the system diverges as

$$N = \frac{1}{\hbar} \int_{x_1}^{x_2} \frac{1}{\pi} \sqrt{2m(\mu - V(u))} du \qquad \text{when} \quad \hbar \to 0. \qquad (9)$$

We are now ready to express the main result of this paper. Sending $\hbar \to 0$ while keeping all other parameters fixed, including the positions $x$ and $x'$ with $x \neq x'$, we have

$$g_1(x, x') = |C|^2 \sqrt{\frac{m}{2\hbar \tilde{L}}} \frac{\left| \sin\left( \frac{\pi}{\tilde{L}} \tilde{x} \right) \right|^{\frac{1}{4}} \left| \sin\left( \frac{\pi}{\tilde{L}} \tilde{x}' \right) \right|^{\frac{1}{4}}}{\left| \sin\left( \frac{\pi}{\tilde{L}} \frac{\tilde{x} - \tilde{x}'}{2} \right) \right|^{\frac{1}{2}} \left| \sin\left( \frac{\pi}{\tilde{L}} \frac{\tilde{x} + \tilde{x}'}{2} \right) \right|^{\frac{1}{2}}} \qquad (10)$$

at the leading order in $\hbar$, whereas higher order corrections vanish in the limit (6). These corrections correspond to finite-size effects and may be significant in real systems with finite $N$, yet they will be investigated elsewhere. The numerical constant $|C|^2$ is the same as above, see Eq. (5). It is an easy exercise, which we leave to the reader, to check that when the potential $V(x)$ is harmonic, Eq. (10) coincides with the result of Refs. [23, 24] (see Table 1).

The rest of the paper is organized as follows. In section 2 we briefly explain how to deal with the calculation of $g_1(x, x')$ in CFT in the case of constant density. The core of the paper is section 3, where we adapt this calculation to deal with the inhomogeneous case, relying on the results of Ref. [35], and where we derive our main formula (10). We provide numerical checks of Eq. (10) in section 4, and conclude in section 5.

## 2 The homogeneous case: solution from mapping on the Gaussian free field

In this section we assume that there is no confining potential $V(x) = 0$, so that the density of particles in the ground state is constant, $\langle \rho(x) \rangle = \rho$.

In order to make use of conformal invariance in 2d, we start by reformulating the one-particle density matrix (3), which is an object in a 1d quantum mechanics problem, as a correlation function in a 2d euclidean field theory. This is done by introducing a time-dependence, and then by performing a Wick rotation $t \rightarrow i\tau$. The two-point function at different positions $x, x'$ and imaginary times $\tau, \tau'$ is defined as

$$\left\langle \Psi^\dagger(x, \tau) \Psi(x', \tau') \right\rangle = \lim_{T \rightarrow \infty} \frac{\langle \psi' | e^{-\frac{(T-\tau)H}{\hbar}} \Psi^\dagger(x) e^{-\frac{(\tau-\tau')H}{\hbar}} \Psi(x') e^{-\frac{(T+\tau')H}{\hbar}} | \psi' \rangle}{\langle \psi' | e^{-\frac{2TH}{\hbar}} | \psi' \rangle}, \qquad (11)$$

where $| \psi' \rangle$ can be any state that has non-zero overlap with the ground state of $H$. The limit $T \rightarrow \infty$ projects onto the ground state, $e^{-\frac{TH}{\hbar}} \sim e^{-\frac{TE_0}{\hbar}} | \psi \rangle \langle \psi |$. The one-particle density matrix (3) is recovered at equal time $\tau = \tau'$.

We view Eq. (11) as a two-point function in some euclidean field theory, to be described in more detail shortly. It is often convenient to work with complex coordinates $z = x + iv\tau$, $\bar{z} = x - iv\tau$, where $v$ is the velocity of gapless excitations of $H$. For an infinitely long system, the coordinate $z$ then lives in the complex plane $\mathbb{C}$, and we shall then sometimes use the notation

$$\left\langle \Psi^\dagger(z, \bar{z}) \Psi(z', \bar{z}') \right\rangle_{\mathbb{C}} \qquad (12)$$

for the correlation function (11).

### 2.1 The fluctuating height field

The Lieb-Liniger model is well-known to be described by Luttinger liquid theory [36], which is equivalent to a gaussian free field (g.f.f.). This is of course textbook material (for instance, a good textbook for our purposes is the one by Tsvelik [37]). In this paragraph, we briefly review this connection, which consists of two main steps. We omit all details that are not directly relevant to the core of this paper.

The first step is to observe that, in one-dimensional space, configurations of indistinguishable particles can be mapped to configurations of a *height function $h(x)$*, see Figure 1. We are mostly interested in the fluctuations of the field $h(x)$—which will turn out to be universal shortly—, as opposed to the ground state expectation value $\langle h(x) \rangle$. Thus, we assume from now on that we have shifted the function $h$,

$$h(x) \rightarrow h(x) - \langle h(x) \rangle, \qquad (13)$$

such that $\langle h(x) \rangle = 0$. The density operator $\rho(x)$ in the initial microscopic model is given by

$$\rho(x) = \rho + \frac{1}{2\pi} \partial_x h(x), \qquad (14)$$

where $\rho$ in the r.h.s is the uniform density in the ground state; the normalization factor $1/(2\pi)$ in front of $\partial_x h$ is introduced for later convenience. Finally, letting the system evolve in imaginary time as in Eq. (11), one gets a fluctuating height field $h(x, \tau)$—or $h(z, \bar{z})$ in complex coordinates—, living in the 2d plane.

The second step is to write an action for the field $h(x, \tau)$. This action must be local, and translation-invariant, because it comes from a microscopic model that has these two properties.

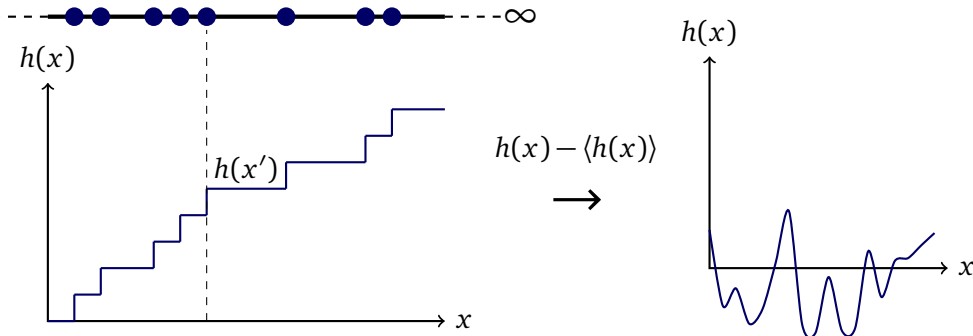

Figure 1: (left) Height function $h(x)$ corresponding to a configuration of particles on the infinite line. (right) By shifting the height function, $h(x) \to h(x) - \langle h(x) \rangle$, one obtains a fluctuating field with zero mean-value.

In addition, since the mapping between particles and the height field is defined only up to a constant shift $h \to h + \text{cst.}$, the action must also possess this symmetry. We then adopt a coarse-grained point of view: one argues that, under the renormalization group (RG) flow, the model should renormalize to a fixed point with scale-invariant action. The only possibility is the gaussian action,

$$S = \frac{1}{2\pi K} \int_{\mathbb{C}} (\partial_z h)(\partial_{\bar{z}} h) \mathrm{d}^2 z, \tag{15}$$

because all other possible terms, including non-quadratic ones, would involve more derivatives and thus be irrelevant. Hence, the field $h$ is a g.f.f. The action is invariant under conformal mapping $z \mapsto f(z)$, as the Jacobian coming from $\mathrm{d}^2 z$ is canceled by the chain rule for the partial derivatives.

The overall normalization factor $K$ in the action (15) cannot be determined by this type of arguments, but let us anticipate slightly and announce that we have in fact

$$K = 1 \tag{16}$$

in the Tonks-Girardeau gas. Then we arrive at the following expression for the propagator,

$$\left\langle h(z,\bar{z}) h(z',\bar{z}') \right\rangle_{\mathbb{C}} = -\log|z - z'|^2, \tag{17}$$

which completely solves the theory. Indeed, all other correlation functions are expressible in terms of this propagator using Wick's theorem.

Now, why $K = 1$? From Eq. (14), we see that the density-density correlation behaves as $\left\langle \rho(x)\rho(x') \right\rangle = \rho^2 - \frac{1}{2\pi^2}|x - x'|^{-2}$. This is consistent with the direct calculation in the microscopic model, which follows from the fact that the Tonks-Girardeau gas maps to free fermions [13]. If $K$ was not equal to one, then the prefactor in the density-density correlation would have been $\frac{1}{2\pi^2} \to \frac{K}{2\pi^2}$, and it would not match the microscopic solution. It is a general fact that, in all free fermion models, the Luttinger parameter $K$ is always equal to one.

## 2.2 Boson creation/annihilation operator

Any local operator in the microscopic model can be written as a linear combination of local operators in the field theory. Those field theory operators can be organized according to their scaling dimension: the lower the scaling dimension, the more important the contribution to correlation functions. In order to get the asymptotic behavior of $g_1(x, x')$ from field theory, our goal is thus to identify the field theory operator with the smallest scaling dimension that contributes to the boson creation operator $\Psi^\dagger$. The local operators in the field theory are the

vertex operators, the operator $\partial h$, and their descendants, that have higher scaling dimensions. Since the operator $\partial h$ has charge zero, it cannot appear in the expansion of $\Psi^\dagger$. Hence, the leading contribution must come from a vertex operator,

$$\Psi^\dagger(z,\bar{z}) = B : e^{i\alpha\varphi(z)+i\beta\overline{\varphi}(\bar{z})} : + \text{less relevant operators}, \tag{18}$$

where $B$ is some constant, and $\alpha$ and $\beta$ will be determined shortly. The fields $\varphi(z)$ and $\overline{\varphi}(\bar{z})$ are the chiral and anti-chiral parts of the height field $h(z,\bar{z})$ (for more details, see for instance Ref. [37]), with the following propagators,

$$h(z,\bar{z}) = \varphi(z) + \overline{\varphi}(\bar{z}), \quad \left\langle \varphi(z)\varphi(z') \right\rangle_{\mathbb{C}} = -\log(z-z'), \quad \left\langle \overline{\varphi}(\bar{z})\overline{\varphi}(\bar{z}') \right\rangle_{\mathbb{C}} = -\log(\bar{z}-\bar{z}'), \tag{19}$$

and $\left\langle \varphi(z)\overline{\varphi}(\bar{z}) \right\rangle_{\mathbb{C}} = 0$. In order for the identification (18) to make sense, we need to check that it is compatible with the relation

$$\left[\rho(x),\Psi^\dagger(x')\right] = \delta(x-x')\Psi^\dagger(x'). \tag{20}$$

To check this, we use Eq. (19) and Wick's theorem to get the operator product expansion

$$\partial_x h(z,\bar{z}) : e^{i\alpha\varphi(z')+i\beta\overline{\varphi}(\bar{z}')} := (\partial_z \varphi(z) + \partial_{\bar{z}} \overline{\varphi}(\bar{z})) : e^{i\alpha\varphi(z')+i\beta\overline{\varphi}(\bar{z}')} :$$
$$\sim \left(-\frac{i\alpha}{z-z'} - \frac{i\beta}{\bar{z}-\bar{z}'}\right) : e^{i\alpha\varphi(z')+i\beta\overline{\varphi}(\bar{z}')} :, \tag{21}$$

and then we evaluate the commutator of the density operator (14) with the vertex operator by playing with the imaginary part of the coordinate $z$,

$$\left[\rho(x), : e^{i\alpha\varphi(x')+i\beta\overline{\varphi}(x')} :\right] = \lim_{\varepsilon\to 0^+} \frac{1}{2\pi} \left[\partial_x h(x+i\varepsilon, x-i\varepsilon) : e^{i\alpha\varphi(x')+i\beta\overline{\varphi}(x')} : -\{\varepsilon \to -\varepsilon\}\right]$$
$$= \lim_{\varepsilon\to 0^+} \frac{1}{2\pi} \left[-\frac{i\alpha}{x-x'+i\varepsilon} - \frac{i\beta}{x-x'-i\varepsilon} - \{\varepsilon \to -\varepsilon\}\right] : e^{i\alpha\varphi(x')+i\beta\overline{\varphi}(x')} :$$
$$= (\beta-\alpha)\delta(x-x') : e^{i\alpha\varphi(x')+i\beta\overline{\varphi}(x')} : . \tag{22}$$

Comparing (20) and (22), we see that $\beta - \alpha = 1$. We thus need to minimize the scaling dimension of the vertex operator $\frac{1}{2}(\alpha^2 + \beta^2)$, while satisfying this constraint, and we easily find

$$\min_{\beta-\alpha=1} \frac{1}{2}(\alpha^2 + \beta^2) = \frac{1}{4}, \qquad \text{for} \quad \beta = -\alpha = \frac{1}{2}. \tag{23}$$

To summarize, we have identified the leading operator that contributes to $\Psi^\dagger$. It is the vertex operator $: e^{-i\frac{1}{2}(\varphi(z)-\overline{\varphi}(\bar{z}))} :$. Now, we focus on the constant $B$ in Eq. (18). The vertex operator has scaling dimension $1/4$, but the boson creation operator has scaling dimension $1/2$ (this is easily seen by looking at the commutation relation of the creation and annihilation operator). By dimensional analysis, the constant $B$ has the dimension of a length to the power $-1/4$. Since the only microscopic length scale in the Tonks-Girardeau gas is the density, $B$ must scale with the density $\rho$. We therefore have $B = C\rho^{\frac{1}{4}}$, where $C$ is a dimensionless constant. We arrive at

$$\Psi^\dagger(x) = C\rho^{\frac{1}{4}} : e^{-i\frac{1}{2}[\varphi(z)-\overline{\varphi}(\bar{z})]} : + \text{less relevant operators}, \tag{24}$$

and the annihilation operator has a similar expansion

$$\Psi(x) = C^* \rho^{\frac{1}{4}} : e^{i\frac{1}{2}[\varphi(z)-\overline{\varphi}(\bar{z})]} : + \dots \tag{25}$$

The constant $C$ cannot be fixed by this type of arguments; it really needs to be extracted from the solution of the microscopic model. Such a microscopic solution is beyond the scope of the present paper, however, for consistency with the results reviewed in the introduction, we know that $|C|$ must be given by Eq. (5). The phase of $C$ is undetermined, which reflects the symmetry of the Hamiltonian (1) under $U(1)$ transformations $\Psi^\dagger \to e^{i\theta}\Psi^\dagger$, $\Psi \to e^{-i\theta}\Psi$.

### 2.3 One-particle density matrix in the homogeneous cases

As mentioned in the beginning of section 2, the coordinates in the homogeneous cases are $z = x + i\nu\tau$ and $z' = x' + i\nu\tau'$, with the one-particle density matrix corresponding to the case $\tau = \tau'$. For the translationally-invariant Tonks-Girardeau gas, that is the one in the first line of Table 1, we have

$$
\begin{aligned}
\left\langle \Psi^\dagger(z,\bar{z})\Psi(z',\bar{z}')\right\rangle_{\mathbb{C}} &= |C|^2 \rho^{\frac{1}{2}} \left\langle : e^{-i\frac{1}{2}[\varphi(z)-\overline{\varphi}(\bar{z})]} :: e^{i\frac{1}{2}[\varphi(z')-\overline{\varphi}(\bar{z}')]} : \right\rangle_{\mathbb{C}} \\
&= |C|^2 \rho^{\frac{1}{2}} \left|z-z'\right|^{-\frac{1}{4}} \left|\bar{z}-\bar{z}'\right|^{-\frac{1}{4}} \\
&\stackrel{\{\tau=\tau'\}}{=} |C|^2 \frac{\rho^{\frac{1}{2}}}{|x-x'|^{\frac{1}{2}}} \; ,
\end{aligned}
\tag{26}
$$

where we have used the formula for the expectation value of vertex operators [37]

$$
\left\langle \prod_{i=1}^{l} : e^{i\alpha_i \varphi(z_i)} : \right\rangle_{\mathbb{C}} = \prod_{1 \le j < k \le l} \left|z_j - z_k\right|^{\alpha_j \alpha_k} \; .
\tag{27}
$$

Using conformal symmetry, we can also easily recover the periodic case, that is the one in the second line of Table 1, for which $x + L$ is identified with $x$. The complex coordinate $z$ now lives on the cylinder $\mathscr{C}$, but is mapped onto the plane $\mathbb{C}$ by the conformal transformation $z \mapsto w(z) = e^{i\frac{2\pi}{L}z}$, where $L$ is the circumference of the cylinder. Hence, the one-particle density matrix is evaluated in the same fashion as in the infinite-line case,

$$
\begin{aligned}
\left\langle \Psi^\dagger(z,\bar{z})\Psi(z',\bar{z}')\right\rangle_{\mathscr{C}} &= |C|^2 \rho^{\frac{1}{2}} \left\langle : e^{-i\frac{1}{2}[\varphi(z)-\overline{\varphi}(\bar{z})]} :: e^{i\frac{1}{2}[\varphi(z')-\overline{\varphi}(\bar{z}')]} : \right\rangle_{\mathscr{C}} \\
&= |C|^2 \rho^{\frac{1}{2}} \left|\frac{dw}{dz}\right|^{\frac{1}{4}} \left|\frac{dw'}{dz'}\right|^{\frac{1}{4}} \left\langle : e^{-i\frac{1}{2}[\varphi(w)-\overline{\varphi}(\bar{w})]} :: e^{i\frac{1}{2}[\varphi(w')-\overline{\varphi}(\bar{w}')]} : \right\rangle_{\mathbb{C}} \\
&\stackrel{\{\tau=\tau'\}}{=} |C|^2 \frac{\rho^{\frac{1}{2}}}{\left|\frac{L}{\pi}\sin\left(\frac{\pi}{L}x-x'\right)\right|^{\frac{1}{2}}} \; .
\end{aligned}
\tag{28}
$$

We just showed that, after identifying the boson creation and annihilation operators with Eqs. (24) and (25), we indeed recover the first two cases of Table 1 for constant density $\rho$.

## 3 The case with a trapping potential

In section 2, we have reviewed the key points allowing a simple computation of the leading behavior of $g_1(x,x')$ in the cases of uniform density. In this section we switch on a confining potential $V(x)$, such that the density is now position dependent $\langle\rho(x)\rangle$. On top of the inhomogeneous density, the second important thing to bear in mind is that we have to deal with boundary conditions. Because of the potential, the particles are confined to an interval $[x_1, x_2]$, outside which the density is 0 as in Eq. (7). So we must ask: what is the boundary condition satisfied by the height function $h(x)$ at the boundaries $x_1$ and $x_2$? With the definition given in Eq. (13), its generalization to finite-size is just one step away.

For a system of width $x_2 - x_1$, the height field can be written explicitly as

$$
h(x) \sim 2\pi \int_{x1}^{x} du \; (\rho(u) - \langle\rho(u)\rangle)
\tag{29}
$$

Thus, since the total number of trapped particles N is fixed, we see that we have $h(x_1) = h(x_2) = 0$, that is Dirichlet boundary condtions. The field operator in the r.h.s of Eq. (24) can now be viewed as living on an infinite vertical strip (see Figure 2). So far so good, but how is inhomogeneity coming up in the theory? The answer is to be found in the action (15).

## 3.1 Conformal field theory in curved 2d space

The main insight from Ref. [35] is that, in the inhomogeneous case the velocity of the excitations is position dependent, which means that the metric appearing in the action (15) is no longer just the flat metric, $ds^2 \neq dz d\bar{z}$. For the theory in curved space to be consistent, correlations involving operators (24) and (25) should exhibit the same local behavior as in the uniform case with the same local density. It was argued in Ref. [35] that the knowledge of the local behavior of the propagator, together with the constraint that the action is local, is sufficient to unravel the field theory action in the inhomogeneous case. This task boils down to finding a system of isothermal coordinates $z(x, \tau)$ and $\bar{z}(x, \tau)$, in which the metric takes the form

$$ds^2 = dx^2 + v^2(x)d\tau^2 = e^{2\sigma(z,\bar{z})}dz d\bar{z} , \tag{30}$$

where the function $\sigma(z, \bar{z})$ is real-valued. A convenient choice for the coordinate $z$ is [35]

$$z(x, \tau) = \int_{x_1}^{x} \frac{du}{v(u)} + i\tau = \tilde{x} + i\tau \qquad \text{with} \qquad e^{\sigma(x)} = v(x), \tag{31}$$

where $v(x)$ is the velocity of gapless excitations given by Eq. (8). As pointed out in the introduction, the variable $\tilde{x}$ has a clear physical interpretation: it is the time that a particle takes to travel from the left boundary $x_1$ to a position $x$. From now on, the strip is parametrized by the coordinates $(\tilde{x}, \tau) \in [0, \tilde{L}] \times \mathbb{R}$, where $\tilde{L}$ is the time needed to travel from one boundary to the other.

We thus need to evaluate the expectation value of operators in the strip $\mathscr{S}_{\text{curved}} = [0, \tilde{L}] \times \mathbb{R}$ equipped with the curved background metric $ds^2 = e^{2\sigma(z,\bar{z})}dz d\bar{z}$. However, by performing the Weyl transformation

$$ds^2 = e^{2\sigma(z,\bar{z})}dz d\bar{z} \rightarrow ds^2 = dz d\bar{z}, \tag{32}$$

we can bring this back to the calculation of a correlation function in the flat strip $\mathscr{S}_{\text{flat}}$, namely $[0, \tilde{L}] \times \mathbb{R}$ with the metric $ds^2 = dz d\bar{z}$. Under the above transformation, a primary operator $\phi(z, \bar{z})$ transforms as $\phi(z, \bar{z}) \rightarrow e^{-\Delta\sigma(z,\bar{z})}\phi(z, \bar{z})$, where $\Delta$ is the scaling dimension of $\phi$. In particular, the vertex operator in Eq. (24) transforms as

$$: e^{-i\frac{1}{2}[\varphi(z)-\overline{\varphi}(\bar{z})]} : \longrightarrow \; e^{-\frac{\sigma(z,\bar{z})}{4}} : e^{-i\frac{1}{2}[\varphi(z)-\overline{\varphi}(\bar{z})]} : . \tag{33}$$

Applying this to the two-point correlator of interest, we get

$$\begin{aligned}
\left\langle \Psi^{\dagger}(z, \bar{z})\Psi(z', \bar{z}') \right\rangle_{\mathscr{S}_{\text{curved}}} &= |C|^2 \left\langle \rho(x) \right\rangle^{\frac{1}{4}} \left\langle \rho(x') \right\rangle^{\frac{1}{4}} \left\langle : e^{-i\frac{1}{2}[\varphi(z)-\overline{\varphi}(\bar{z})]} :: e^{i\frac{1}{2}[\varphi(z')-\overline{\varphi}(\bar{z}')]} : \right\rangle_{\mathscr{S}_{\text{curved}}} \\
&= |C|^2 \left( \frac{\langle\rho(x)\rangle}{e^{\sigma(x)}} \right)^{\frac{1}{4}} \left( \frac{\langle\rho(x')\rangle}{e^{\sigma(x')}} \right)^{\frac{1}{4}} \left\langle : e^{-i\frac{1}{2}[\varphi(z)-\overline{\varphi}(\bar{z})]} :: e^{i\frac{1}{2}[\varphi(z')-\overline{\varphi}(\bar{z}')]} : \right\rangle_{\mathscr{S}_{\text{flat}}}.
\end{aligned} \tag{34}$$

Here, we again used the expressions (24) and (25), but we substituted the constant density $\rho$ for the local density in Eq. (7), $\rho \rightarrow \langle\rho(x)\rangle$. Finally, the terms $\left( \frac{\langle\rho(x)\rangle}{e^{\sigma(x)}} \right)^{\frac{1}{4}}$ and $\left( \frac{\langle\rho(x')\rangle}{e^{\sigma(x')}} \right)^{\frac{1}{4}}$ merely give the constant term $\sqrt{\frac{m}{\pi\hbar}}$, as can be seen by looking at Eqs. (8) and (31). Thus, the remaining task is simply to evaluate the two-point function on $\mathscr{S}_{\text{flat}}$ in the r.h.s of Eq. (34).

## 3.2 Mapping on the upper-half-plane: method of images

To calculate the two-point function on $\mathscr{S}_{\text{flat}}$ in Eq. (34), we use a conformal mapping, like in section 2.3. The difference here though, is that we now have a boundary. To deal with boundaries in conformal field theory, the common trick is to use the method of images, which comes

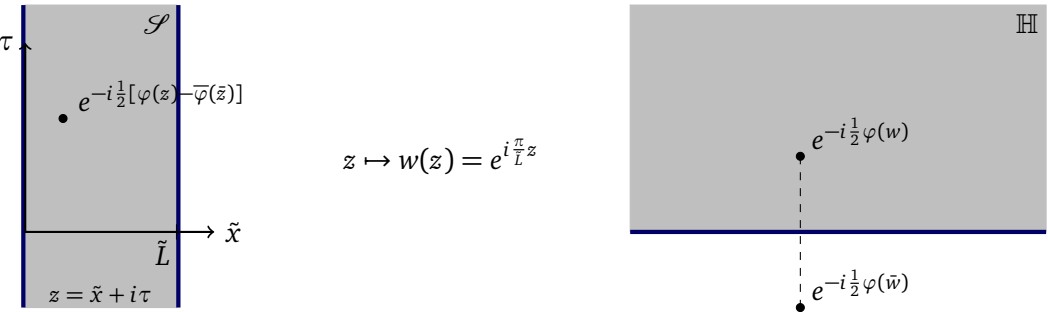

Figure 2: Infinite strip $\mathscr{S} = [0, \tilde{L}] \times \mathbb{R}$. It is mapped to the upper-half-plane $\mathbb{H}$ by the conformal transformation $z \mapsto w(z) = e^{i \frac{\pi}{\tilde{L}} z}$. It allows to exactly compute the normal ordered product on the plane $\mathbb{C}$ using the method of images.

from electrostatics [38]. Essentially, it consists in rewriting an $n$-point correlation function in a domain with a boundary as a $2n$-point correlation function in the plane $\mathbb{C}$ [39]. One starts by mapping the strip onto the upper-half-plane $\mathbb{H}$ with the transformation $z \mapsto w(z) = e^{i \frac{\pi}{\tilde{L}} z}$, such that the correlator can be put in the form

$$\left\langle \Psi^{\dagger}(z, \bar{z}) \Psi(z', \bar{z}') \right\rangle_{\mathscr{S}_{\text{curved}}} = |C|^2 \sqrt{\frac{m}{\pi \hbar}} \left| \frac{dw}{dz} \right|^{\frac{1}{4}} \left| \frac{dw'}{dz'} \right|^{\frac{1}{4}} \left\langle : e^{-i \frac{1}{2} [\varphi(w) - \overline{\varphi}(\bar{w})]} :: e^{i \frac{1}{2} [\varphi(w') - \overline{\varphi}(\bar{w}')]} : \right\rangle_{\mathbb{H}}.$$

In the upper-half-plane $\mathbb{H}$, we have a Dirichlet boundary condition along the real axis, which reads $h(w, \bar{w}) = \varphi(w) + \overline{\varphi}(\bar{w}) = 0$ if $\text{Im}(w) = 0$. This means that the fields $\varphi$ and $\overline{\varphi}$ are no longer independent, since we have $\varphi = -\overline{\varphi}$ along the real axis. One can then view $\overline{\varphi}$ as the analytic continuation of $-\varphi$ to the lower half-plane, so that the operator $: e^{-i \frac{1}{2} \overline{\varphi}(\bar{w})} :$ can be replaced by $: e^{i \frac{1}{2} \varphi(\bar{w})} :$ in the above expression, see Fig. 2. Thus, it allows to express the two-point function in the upper-half-plane $\mathbb{H}$ as a four-point function in the plane $\mathbb{C}$,

$$\left\langle : e^{-i \frac{1}{2} [\varphi(w) - \overline{\varphi}(\bar{w})]} :: e^{i \frac{1}{2} [\varphi(w') - \overline{\varphi}(\bar{w}')]} : \right\rangle_{\mathbb{H}} = \left\langle : e^{-i \frac{1}{2} \varphi(w)} :: e^{-i \frac{1}{2} \varphi(\bar{w})} :: e^{i \frac{1}{2} \varphi(w')} :: e^{i \frac{1}{2} \varphi(\bar{w}')} : \right\rangle_{\mathbb{C}}.$$

The rest of the calculation is straightforward. We use formula (27) once more, and we get

$$\left\langle \Psi^{\dagger}(z, \bar{z}) \Psi(z', \bar{z}') \right\rangle_{\mathscr{S}_{\text{curved}}} = |C|^2 \sqrt{\frac{m}{\pi \hbar}} \left| \frac{dw}{dz} \right|^{\frac{1}{4}} \left| \frac{dw'}{dz'} \right|^{\frac{1}{4}} \frac{|w - \bar{w}|^{\frac{1}{4}} |w' - \bar{w}'|^{\frac{1}{4}}}{|w - w'|^{\frac{1}{2}} |w - \bar{w}'|^{\frac{1}{2}}}. \tag{35}$$

The last step consists in rewriting Eq. (35) in terms of the original coordinates $z$ and $\bar{z}$. When specializing to equal time $\tau = \tau'$, this gives our main result (10).

## 4 Numerical checks

In principle, the one-particle density matrix of the Tonks-Girardeau gas can be tackled using free fermion techniques; one has to solve numerically the Schrödinger equation for an arbitrary potential and then compute determinants of some large matrices. There exist very efficient algorithms based on the lattice version of this problem [40], which map to the continuum problem by working at low fillings [41]. Nonetheless, in order to treat numerically this problem, we chose to use a different method. We also rely on the fact that the spinless free fermion gas is equivalent to the continuum limit of the quantum XX spin-$\frac{1}{2}$ chain, which allows us to deal with a discrete lattice model in place of a continuous one. However, instead of calculating

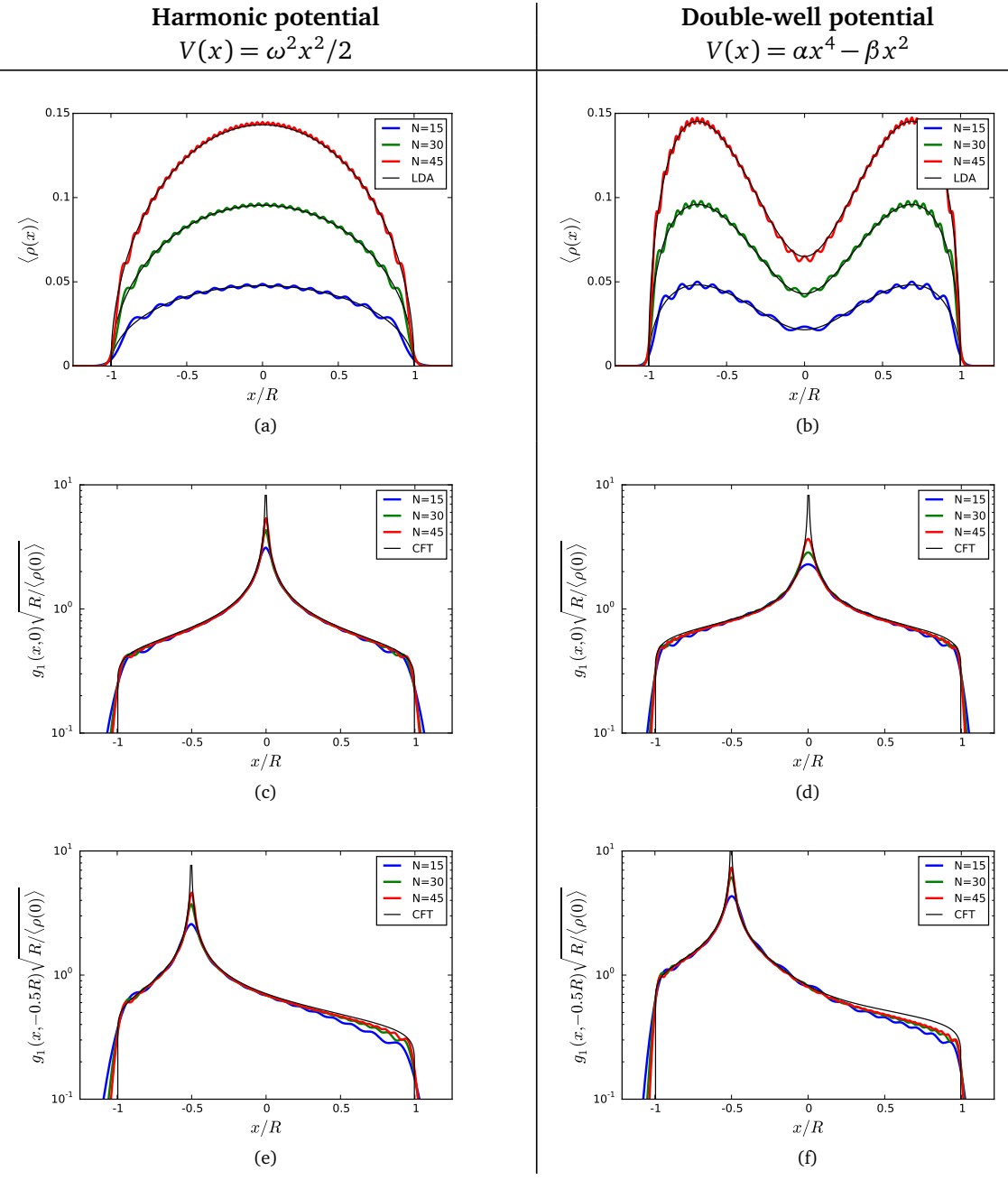

Table 2: DMRG computations in the XX chain with low density of particles, used as an approximation to the continuous Tonks-Girardeau gas. In all the figures, the number of trapped particles is denoted by $N$. The top row shows the local density $\langle \rho(x) \rangle$ for both the harmonic potential trap (a) and the double-well potential (b), which serves to check that the LDA is applicable. In the next two rows the correlations $g_1(x, x')$ are plot against $x/R$, for $x' = 0$ in plots (c) and (d), and $x' = -0.5R$ in plots (e) and (f). The correlations are normalized by $\sqrt{R/\langle \rho(0) \rangle}$.

correlation functions in that spin chain by numerical evaluation of some large determinants (using the results of Ref. [42]), we find that it is just as convenient to use a modern DMRG (density matrix renormalisation group) library such as the code available in the open-source C++ library ITensor [43].

The Hamiltonian of the XX chain in an arbitrary trapping potential $V(x)$ reads

$$H = -\frac{J}{2} \sum_{j=1}^{\mathcal{N}-1} S_j^+ S_{j+1}^- + S_j^- S_{j+1}^+ + \sum_{j=1}^{\mathcal{N}} (J - \mu + V(ja_0)) S_j^z, \tag{36}$$

where $x = ja_0$, with $j \in \mathbb{N}$ and $a_0$ the lattice spacing, $S_j^{\pm,z}$ are the Pauli matrices at site $j$, $\mathcal{N}$ stands for the number of sites, $\mu$ is the chemical potential and $J = \hbar^2/ma_0^2$, with $m$ the mass of a particle. First, we need to take the continuum limit of the XX chain. To do so, we know that the Hamiltonian (36) is mapped to spinless free fermions on the lattice by a Jordan-Wigner transformation. When $V(x) = 0$, this spinless free fermion Hamiltonian is diagonal in momentum space, here denoted by $k$, and yields the kinetic term $J(1 - \cos(ka_0)) - \mu$. Hence, the continuum limit is given by the low-density regime, $a_0 \ll \rho_{max}^{-1} \sim k^{-1}$, where $\rho_{max}$ is the maximal density in the trap, and switching on the confining potential $V(x)$ we obtain a semiclassical picture for the energy of a particle with momentum $k$ at position $x$,

$$E = \frac{\hbar^2 k^2}{2ma_0^2} - \mu + V(x). \tag{37}$$

Now, for the local density approximation to hold, the variation of the density must be smooth on the scale of the interparticle distance, that is $\langle \rho(x) \rangle \gg \frac{|\partial_x \langle \rho(x) \rangle|}{\langle \rho(x) \rangle}$. Since we fix $m$, $a_0$, $\mu$ and $V(x)$, the size of the trap is also fixed and we can introduce the shorthand notation $2R = x_2 - x_1$, with $[x_1, x_2]$ the interval in which $\mu - V(x) > 0$, see Eqs. (7) and (9). Again, the total number of particles $N$ is tuned by varying $\hbar$. With this at hand, we have $\frac{|\partial_x \langle \rho(x) \rangle|}{\langle \rho(x) \rangle} \propto \frac{1}{R}$ and the above condition is equivalent to taking $R$ much larger than $\rho_{max}^{-1}$.

Putting everything together, we must impose the following condition

$$a_0 \ll \rho_{max}^{-1} \ll 2R \lesssim \mathcal{N} a_0. \tag{38}$$

If this scale seperation holds, then Eq. (37) is verified and the density profile is given by the LDA (7). The one-particle density matrix (3) can now be identified with the spin-spin correlation function in the ground state of the Hamiltonian (36),

$$g_1(x, x') = \langle \psi | S_j^+ S_{j'}^- | \psi \rangle, \qquad \text{with } x = ja_0. \tag{39}$$

The results are gathered in Table 2. In all the computations, the total number of sites is $\mathcal{N} = 500$ and $m = a_0 = 1$, so that $x = j$ and $\hbar = \sqrt{J}$. For the reader's information, we here give the set of parameters chosen in the double-well potential case, satisfying the scale separation (38): for $\alpha = 5 \times 10^{-13}$, $\beta = 2 \times 10^{-8}$ and $\mu = 5 \times 10^{-5}$, the size of the trap is $R \simeq 200$, and the number of trapped particles $N = 15, 30, 45$ corresponds to $J = 0.0218, 0,0055, 0,0024$ respectively. The agreement with the CFT prediction is fairly good already for $N = 15$ particles, and improves when $N$ increases.

## 5 Conclusion

The main result (10) and the motivation of this paper were summarized in the introduction. We would like to conclude by discussing possible extensions of this result.

First, let us emphasize that the technique we used, which was developed in Ref. [35], is not specific to the one-particle density matrix $g_1(x, x')$. As long as the limit (6) holds, this approach is very powerful and can be used to calculate the exact asymptotic behavior of any other ground state correlation function of local operators in the trapped Tonks-Girardeau gas.

Second, one may wonder whether our treatment can be applied to the 1d delta Bose gas with finite interaction strength $g$—see the Hamiltonian (1)—. The inhomogeneous case is a problem of great interest in cold atoms, see e.g. [9, 21, 44–49], and it should be noted that, even in the homogeneous case ($V(x) = 0$) the calculation of correlation functions directly from the microscopic model is a hard task [50–52]. We believe that the approach we followed here should be amenable to calculations also at finite interaction strength $g$. However, to see what would be different in this case, recall that, in Luttinger liquid theory, there are two parameters: the velocity of gapless excitations $v$ and the Luttinger parameter $K$. In inhomogeneous situations, the velocity becomes position dependent, $v \to v(x)$, and we expect that $K$ generically does the same, $K \to K(x)$. This does not fundamentally change the approach followed here, but it quickly becomes more technical. We hope to come back to this interesting problem in subsequent work.

In contrast, in problems where the Luttinger parameter $K$ remains constant, we expect that our calculation should generalize straightforwardly. One example where this should happen is the closely related problem of impenetrable anyons, which has been thoroughly studied in the past decade, see e.g. Refs. [53–58]. In particular, our result (10) is to be linked with recent results obtained in Ref. [58], where the authors derive the one-body density matrix of impenetrable anyons in a harmonic trap, generalizing the approach followed by Gangardt [24]. Indeed, the key point is that impenetrable anyons correspond to a Luttinger liquid (or gaussian free field) with a Luttinger parameter $K$ that is fixed by the anyons' statistics only, and is therefore independent of the density. Thus, just like in the $K = 1$ case which we analyzed here, the Luttinger parameter is always a constant in these models, and only $v(x)$ varies with position. The calculation is therefore of the same type as the one we did in this paper.

# Acknowledgements

We thank Pasquale Calabrese, Jean-Marie Stéphan and Jacopo Viti for joint work on Ref. [35], which served as the starting point for the present project, as well as for their critical reading of the manuscript and their important comments. We also acknowledge stimulating discussions with Tom Claeys and Karol Kozlowski. We thank Christophe Chatelain for his help with the ITensor library and Marcos Rigol for inspiring correspondence and sharing his unpublished data. JD thanks the Université Catholique de Louvain, YB is grateful to the Galileo Galilei Insitute in Florence (during the "Lectures on Statistical Field Theories 2017") for financial support, and YB and JD thank the Institut d'Études Scientifiques de Cargèse (during the school "Quantum integrable systems, CFTs and stochastic processes"), for kind hospitality.

**Funding information.** JD acknowledges support from the CNRS "Défi Inphinity" and from the Région Lorraine and the Université de Lorraine.

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
