# Peer review of "One-particle density matrix of trapped one-dimensional impenetrable bosons from conformal invariance"

_SciPost Physics, doi:SciPost Phys. 2, 012 (2017)_

## Round 1 · Referee Report · Anonymous (Referee 1) · 2017-1-31

Strengths

1, Interesting and new results relevant to experiments.

  1. Clear and pedagogical style.

Weaknesses

I would appreciate a discussion of the applicability domain of the results, see report

Report

The manuscripts presents derivation of one-body density matrix of impenetrable (Tonks-Girardeau) bosons in a system with inhomogeneous density. The method used is Conformal Field Theory adapted for inhomogeneous backgrounds. The result is anticipated and reduces to known expression in all particular cases considered. The manuscript is very clearly written and provides sufficient details which can be followed easily. Overall I recommend the manuscript for publication in SciPost. I would appreciate if the authors can clarify the following points:

  1. The use of Conformal Field Theory for interacting systems of bosons can be used for any interaction strength (in a homogeneous setup). The author should state why their results cannot be applied to any value of interactions. I guess this is related to the density dependence of critical exponents depending on Luttinger parameter K.

  2. The density vanishing at the boundaries of the interval makes the Conformal Field approach invalid there. The domain of validity of the derived results should be stated carefully.

Requested changes

See report

  • validity: top
  • significance: high
  • originality: high
  • clarity: top
  • formatting: perfect
  • grammar: excellent

Author:  Yannis Brun  on 2017-03-08  [id 106]

(in reply to Report 1 on 2017-01-31)
Category:
answer to question

We thank the referee for their thorough report on the manuscript. Below we give answers to the two issues raised in the report.

  1. This is actually explained in the second paragraph of the conclusion. In inhomogeneous situations, the Luttinger parameter $K$ becomes space dependent, that is we have $K(x)$. This does not change dramatically the problem and the approach we followed is expected to generalise to such cases without introducing new concepts. It is more technical though and it is still work in progress. We hope to come up with this generalisation in a future paper.

  2. The elements to address this remark are mentioned in the text but we admit that it could have been clearer. The key to the validity of our results is the Local Density Approximation (LDA), which serves in the derivation of all the quantities of interest (group velocity, density-dependent prefactor in the bosonic creation/destruction operators, etc...). In the problem we treat, the thermodynamic limit is taken by sending $\hbar \to 0$ while keeping all other parameters fixed; this is equivalent to sending $N \to \infty$, and it ensures that LDA becomes exact. In other words, we consider the one-particle density matrix $g(x,x')$ with $x$ and $x'$ fixed (or equivalently $x/L$ and $x'/L$ fixed, where $L=x_2-x_1$ is the system size) and send $\hbar \to \infty$, so the points $x$ and $x'$ should always be viewed as bulk points, in the sense that there is always an arbitrarily large number of particles between the position $x$ and the boundary of the system. In this limit, our results are exact. Now, if we were considering small but finite $\hbar$, or equivalently large but finite $N$, there would be subleading corrections. Such corrections are very interesting, but they are beyond the scope of this paper, and will be studied elsewhere. We reformulated the fifth paragraph of the introduction to make the domain of validity of our results more apparent.

---

## Round 1 · Referee Report · Anonymous (Referee 2) · 2017-2-21

Strengths

  1. Very well written, with the aim to reach a broad audience.

Weaknesses

See report

Report

In this paper the authors analyse the equilibrium properties of a one-dimensional Bose gas in the strong repulsive interacting regime. In particular, they focus on the two-point bosonic correlation function, solving the problem of finding its shape in the presence of an arbitrary confining potential by exploiting a very useful relation with a two-dimensional conformal field theory in curved space. Such approach was recently introduced by one of the author (see Ref. 34 and 35) in the context of noninteracting fermions, and here it is further developed so as to obtain a very attractive result for a Tonks gas. I really appreciated this work and I think it absolutely deserves to be published on SciPost.
In spite of that, I have a curiosity that the author may address:

  1. In the present work the author consider hard-core bosons. In order to evaluate the two-point function, they first refer to a Luttinger Liquid gaussian action (Eq.15) with depends on the Luttinger parameter K (which for hard-core bosons is identically 1); thereafter they identify the vertex operators which mostly contribute to the original boson creation operator. Then they use the gaussian action with a properly deformed new metric. Now, I was wondering if this approach can be extended to a generic critical system whose low energy description is given by the Luttigner Liquid theory. For example, in the XXZ spin-1/2 chain with space-dependent coupling \Delta(x) in [-1,1] and space-dependent potential V(x) coupled to Sz (nothing but the density). In this case I may imagine to have a Luttinger parameter K(\rho(x), \Delta(x)) depending on space via the filling and the interaction. Is the author approach effective in this situation or maybe there are some subtleties which make that such approach fails?

Requested changes

See report

  • validity: high
  • significance: high
  • originality: good
  • clarity: top
  • formatting: excellent
  • grammar: good

Author:  Yannis Brun  on 2017-03-08  [id 107]

(in reply to Report 2 on 2017-02-21)
Category:
answer to question

We are grateful to the referee for their positive feedbacks on the manuscript.
Here, we want to briefly discuss the "curiosity" mentioned in the report. First of all, the approach applies to any system captured by Luttinger Liquid theory, whether it is on the lattice or in the continuum. What matters is that the system under scrutiny can be treated with LDA. In the paper we focused on the Tonks-Girardeau gas because it is a particularly simple and physically relevant model, but we clearly think that the method we used can be generalized. As for the specific example of the XXZ chain with space-dependent couplings (that would vary slowly at the lattice scale, in order for LDA to be applicable), it all boils down to introducing space-dependence in the Luttinger paramter $K(x)$. Thus, our answer is the same as the one to the first remark of the previous report: it is more technical but the approach can be generalized to that case. We hope to come back to this in the near future.

---

## Round 2 · Author Response

We thank the editor for taking in charge the submission and refereeing of our manuscript. We here resubmit a new version of the manuscript with the following changes:

---

## Round 2 · List of Changes

- we added references and acknowledgements
- we corrected mistakes in the legend for the plots of Table 2.
- as was asked in the Anonymous Report 74 (on 2017-01-31): we reformulared the fifth paragraph of the introduction so that the domain of validity of the result is more apparent.
- we fixed some typos

---

## Editorial Decision

published